# Two-Dimensional Sb Modified TiO_2_ Nanorod Arrays as Photoanodes for Efficient Solar Water Splitting

**DOI:** 10.3390/nano13071293

**Published:** 2023-04-06

**Authors:** Jie Gao, Shengqi Zhang, Xiaoqing Ma, Yi Sun, Xiaoyan Zhang

**Affiliations:** 1Department of Chemistry, College of Sciences, Shanghai University, Shanghai 200444, China; 2School of Materials Science and Engineering, Shanghai University of Engineering Science, Shanghai 201620, China; 3Aerospace Hydrogen Energy (Shanghai) Technology Co., Ltd., Shanghai 200241, China; 4Power-Sources of Space-Sources Technology, Shanghai Institute of Space State Key Laboratory, Shanghai 200233, China

**Keywords:** TiO_2_ nanorod arrays, antimonene, heterojunction, photoelectrochemical, solar water splitting

## Abstract

As one of the widely studied semiconductor materials, titanium dioxide (TiO_2_) exhibits high photoelectrochemical (PEC) water-splitting performance as well as high chemical and photo stability. However, limited by a wide band gap and fast electron-hole recombination rate, the low solar-to-hydrogen conversion efficiency remains a bottleneck for the practical application of TiO_2_-based photoelectrodes. To improve the charge separation and water oxidation efficiency of TiO_2_ photoanodes, antimonene, a two-dimensional (2D) material obtained by liquid-phase exfoliation, was assembled onto TiO_2_ nanorod arrays (TNRAs) by a simple drop-coating assembly process. PEC measurements showed that the resulting 2D Sb/TiO_2_ photoelectrode displayed an enhanced photocurrent density of about 1.32 mA cm^−2^ in 1.0 M KOH at 0.3 V vs. Hg/HgO, which is ~1.65 times higher than that of the pristine TNRAs. Through UV-Vis absorption and electrochemical impedance spectroscopy measurements, it was possible to ascribe the enhanced PEC performances of the 2D Sb/TiO_2_ photoanode to increased absorption intensity in the visible light region, and improved interfacial charge-transfer kinetics in the 2D Sb/TiO_2_ heterojunction, which promotes electron-hole separation, transfer, and collection.

## 1. Introduction

The rapid development of the global economy and rapid growth in the global population has accelerated the consumption of non-renewable energy. As the reserves of traditional energy are limited, the development of low-cost, clean, and renewable energy is of great significance to ease the energy crisis and the realization of sustainable development. Solar energy, as one of the natural resources on which human beings depend for survival, has attracted great attention because it is inexhaustible. Photoelectrochemical (PEC) water splitting is considered to be one of the most efficient and clean solar energy conversion technologies [1,2,3,4,5], but how to improve the solar energy conversion efficiency of PEC remains a huge challenge.

In a typical PEC water-splitting system, the water oxidation reaction on the photoanode is four orders of magnitude slower than the water reduction reaction on the photocathode, which is the rate-determining step of water splitting [6,7]. How to improve the water decomposition performance of a semiconductor photoanode is the key problem in this field [8]. Semiconductor photocatalysts, especially metal oxides such as TiO_2_, CdS [9], and BiVO_4_ [10], have been widely studied as photoanode materials for PEC. As one of the most widely studied photoanode materials, TiO_2_ [11,12,13] shows good chemical and photo stability, abundant storage capacity, and high photoelectrocatalytic activity. In recent years, researchers have designed different nanostructured TiO_2_ thin films as photoanodes in a PEC cell [14], such as nanorods [15], nanowires [16], nanosheets [17], nanotubes [18], nanoparticles, etc. In particular, one-dimensional (1D) TiO_2_ nanorod arrays (NRAs) show unique morphology and ordered structure that enable charges to migrate rapidly along the axial direction of the NRAs, which significantly improves charge transfer by reducing grain boundaries [19,20]. Meanwhile, the larger specific surface area of 1D TiO_2_ NRAs ensures more active sites on the electrode surface and faster photocarrier migration. However, TiO_2_ can only respond to ultraviolet (UV) light, which accounts for only 3%–5% of sunlight. The photogenerated carriers show serious recombination during the separation and transfer process, resulting in a very low utilization rate of solar energy by TiO_2_. Therefore, various modification strategies to improve the photoresponse and charge separation rate of TiO_2_ NRAs have been reported [21]. Among those strategies, the construction of heterojunctions is one of the most efficient strategies [22]. For example, Wang et al. successfully fabricated g-C_3_N_4_/TiO_2_ nanotube arrays (NTAs) heterojunctions and the new heterostructures of g-C_3_N_4_/TiO_2_ NTAs exhibit enhanced PEC water-splitting activity. The photocurrent density was approximately 2 times (0.86 mA cm^−2^) higher than that of the pristine TiO_2_ NTAs [23]. Liu et al. reported 2D ZnIn_2_S_4_ nanosheet/1D TiO_2_ NR heterojunction arrays synthesized by a facile hydrothermal process. The heterojunction arrays show high performance for solar water splitting as photoelectrodes in a PEC cell [24]. Xie et al. constructed TiO_2_ NRAs/CdS quantum dots/ALD-TiO_2_ nanostructures through atomic layer deposition (ALD), which improved the PEC and photocatalytic properties of TiO2 in solar energy conversion [25].

Recently, 2D antimonene (Sb) nanosheets have attracted much attention because of their layered structure similar to graphene and some unique physical and photoelectric properties superior to graphene [26,27,28,29], such as high carrier mobility, high stability, adjustable band gap, short out-of-plane atom bond length, and strong spin-orbit coupling effect [30,31,32], showing potential applications in the energy storage [33], battery development, optoelectronics [34,35], and semiconductor [36,37,38] fields. The theoretical calculation results from Prof. Zeng’s research group in 2015 show that Sb in the bulk phase has a typical semi-metal band structure, and when it is thinned to a single atomic layer, it will transform into an indirect band-gap semiconductor material [39]. Zhang et al. investigated the atomic and electronic structures of a 2D Sb monolayer using DFT calculations, indicating that 2D Sb with a tunable bandgap is suitable for photocatalytic water splitting [40]. At present, the main methods for the preparation of monolayer and few-layer (<10 layers) Sb nanomaterials are mechanical exfoliation [41], liquid-phase exfoliation [42], and epitaxy growth [43]. In 2018, Wang et al. [26] prepared few-layer Sb nanosheets with smooth surfaces by combining pre-grinding and liquid phase ultrasonic assistance and found that the 2D Sb materials can act as a hole transport layer in perovskite solar cells. Fu et al. [44] synthesized Sb/g-C_3_N_4_ van der Waals heterostructures by self-assembly of g-C_3_N_4_ nanosheets and antimonene that show a better-photogenerated charge separation efficiency than pure g-C_3_N_4_, providing a reference for the rational construction of high-performance heterojunction photocatalyst for CO_2_ reduction reactions. In addition, Zhao et al. [45] reported efficient visible-light N_2_ fixation in water using a novel 2D Sb/amorphous TiO_2_ heterostructure. The intimate interfacial contact of 2D Sb and a-TiO_2_ greatly promoted carrier migration and charge-carrier separation. These findings show the potential application of 2D Sb and its hybrids in photocatalysis. Therefore, it can be expected that Sb nanosheets will have potential applications in PEC fields.

In this work, 2D Sb nanosheets stripped by liquid-phase exfoliation were assembled onto TiO_2_ NRAs by a simple drop-coating assembly process to form heterostructures to improve the PEC performances of TiO_2_ photoanodes. The PEC water splitting results show that the photocurrent density of 2D Sb/TiO_2_ NRAs is 1.32 mA cm^−2^ in 1.0 M KOH at 1.23 V vs. RHE, which is 1.65 times higher than that of the pristine TiO_2_ photoanode. The enhancement can be ascribed to the increased absorption of visible light and separation efficiency of photogenerated electron-hole pairs caused by the introduction of 2D Sb and the formation of heterostructures between 2D Sb and TiO_2_ NRAs. 

## 2. Experimental Section

### 2.1. Materials

Ethanol (>99.8%) and acetone (>99.5%) were purchased from Shanghai Reagent Factory. Fluorine-doped tin oxide conducting glass (FTO, Guluo Glass, LA, CHN) was cleaned by sonication in a mixture of acetone and ethanol and then deionized water for 15 min, respectively, for use. Conductive silver glue was purchased from Shenggelu Technology.

### 2.2. Preparation of 2D Sb Suspensions 

2D Sb dispersions [46,47,48] were prepared by adding 200 mg of bulk antimony powders into 10 mL of N-methyl pyrrolidone (NMP) and sonicating using a horn probe sonic tip (KQ2200V, 100 W) with 40% amplitude for 48 h. The obtained black Sb suspension was then centrifuged at 3000 rpm for 4 min to remove the unshed crystals. The gray-black supernatant containing Sb nanosheets was centrifuged at 12,000 rpm for 12 min. The precipitate was washed three times with absolute ethanol and dried at 60 °C in a vacuum oven. Then, 2, 4, and 6 mg of the obtained 2D Sb were re-dispersed in 1 mL of absolute ethanol under ultrasonication for 30 min to obtain 2D Sb suspensions with concentrations of 2, 4, and 6 mg mL^−1^, respectively. 

### 2.3. Fabrication of 2D Sb Modified TiO_2_ NRAs

TiO_2_ NRAs were grown on FTO substrates via a previously reported hydrothermal method [49,50]. Typically, 15 mL of deionized water was mixed with 15 mL of concentrated hydrochloric acid (36.5–38 wt%). This mixture was stirred for 5 min, then 0.5 mL of titanium butoxide was added dropwise under continuous stirring. The resulting solution was transferred into a Teflon-lined stainless steel autoclave. Then the cleaned FTO conductive glass (1.5 cm × 1.5 cm) was placed vertically in the reaction reactor. The hydrothermal growth was performed at 150 °C for 4 h [51,52]. After cooling to room temperature, the FTO substrate was withdrawn, rinsed extensively with deionized water, and dried at 60 °C in air. The obtained TiO_2_ NRAs were then annealed in air at 400 °C for 1 h. The heterostructured 2D Sb/TiO_2_ NRAs (2D Sb/TNRAs) were prepared by a drop-coating method. Before the drop-coating process, the annealed TiO_2_ NRAs were preheated on a heating plate at 70 °C for 30 min. The obtained 2D Sb suspension of various concentrations (2, 4, or 6 mg mL^−1^) was quickly pipetted and dropped onto the preheated TiO_2_ NRAs with an exposed surface area of 1.5 cm^2^, to make close contact between Sb nanosheets and TiO_2_ NRAs. The final samples were named 2D Sb (2 mg)/TNRAs, 2D Sb (4 mg)/TNRAs, and 2D Sb (6 mg)/TNRAs, based on the concentration of 2D Sb. The electrodes were prepared by connecting FTO/2D Sb/TNRAs with copper wire using conductive silver glue and heating at 60 °C in a vacuum furnace for 2 h. The electrode surface was then coated with insulating silicone rubber, with only a 0.5 cm × 0.6 cm working area left exposed. The obtained electrode was then dried for use.

### 2.4. Characterizations 

X-ray diffraction (XRD) (3 KW D/MAX-2200V/PC) patterns were acquired to analyze the phase and composition of the samples using Cu K*α* radiation (*λ* = 1.5418 Å) with a cathode voltage and current of 30 kV and 30 mA, respectively. Raman spectra were acquired on a LabRAM ARAMIS Raman system using a 532 nm laser as the excitation source. The morphology of the samples was observed under a JEOL microscope (JSM6701F) using scanning electron microscopy (SEM), transmission electron microscopy (TEM), and high-resolution TEM (HRTEM) (FEI-Talos F200S). The chemical states of the samples were characterized by X-ray photoelectron spectroscopy (XPS, ThermoScientific, ESCALAB, 250XI) using monochromatic Al K*α* Radiation. Ultraviolet-visible (UV-Vis) absorption spectra of the obtained samples were recorded with a UV-Vis spectrophotometer (UV-2500PC, Shimadzu) in the range of 300 to 800 nm.

### 2.5. PEC Measurements

All PEC tests were performed using a three-electrode system on a CHI660E electrochemical analyzer under AM 1.5 G (100 mW/cm^2^) illumination, which was produced from a Xenon lamp (300 W) with an AM 1.5 G filter. The PEC performance was measured in 1 M KOH with Hg/HgO as the reference electrode and Pt sheet as the counter electrode. Before the PEC measurements, the system was purged with argon (Ar) for 30 min to remove the oxygen. The potential vs. a reversible hydrogen electrode (RHE) was obtained from the equation: *E*_RHE_ = *E* + 0.059pH + *E*_Hg/HgO_, where *E* is the applied potential vs. Hg/HgO electrode, and *E*_Hg/HgO_ is 0.095 V. Electrochemical impedance spectroscopy (EIS) measurements were made under AM 1.5 G illumination with a frequency range between 0.01 Hz and 100 kHz and an amplitude of 5 mV at the open-circuit potential of the PEC cells.

## 3. Results and Discussion

### 3.1. Morphology and Composition Characterization of 2D Sb/TiO_2_ NRAs

The surface morphology and microstructure of the TiO_2_ NRAs grown on FTO substrates were analyzed by SEM. Figure 1a shows the SEM image of the nanorod arrays. It can be observed that the entire surface of the FTO substrate is covered uniformly with TiO_2_ NRAs. Figure 1b reveals that the TiO_2_ NRAs are tetragonal in shape and their diameters range from 75 to 100 nm with an average length of ~784 nm. After loading 2D Sb onto the TiO_2_ NRAs, the lamellar structure of the 2D Sb nanosheets was observed on the surfaces of the TiO_2_ NRAs (Figure 1c,d), suggesting the successful loading of Sb nanosheets onto the TiO_2_ NRAs. The average thickness of the nanosheets is about 100–200 nm, as shown in Figure 1d. 

The composition and phase structure of the TiO_2_ NRAs and 2D Sb/TNRAs were studied by XRD. As shown in Figure 2a, two sharp reflection peaks at 36.1° and 62.8° marked with black squares are observed for pristine TiO_2_ NRAs, which correspond to the (101) and (002) planes of rutile TiO_2_ (JCPDS No. 88-1175) [53], respectively. The results suggest that the synthesized TiO_2_ NRAs are mainly composed of rutile TiO_2_ as reported previously. For the nanocomposites of 2D Sb/TNRAs, the characteristic diffraction peak of 2D Sb at 28.7° is observed, which is consistent with the previous report [54]. Additionally, a weak peak appeared at 23.7°, corresponding to the (003) plane of bulk antimony (JCPDS No. 35-0732). No characteristic diffraction peaks ascribed to antimonene oxide are observed, suggesting that the Sb in 2D Sb/TNRAs remained unoxidized [52]. The morphologies and crystal structure of 2D Sb were observed by transmission electron microscopy (TEM) and high-resolution transmission electron microscopy (HRTEM). The layered structure of the 2D Sb nanosheet is clearly observed in Figure 2b. Furthermore, a lattice fringe of 0.36 nm is observed in Figure 2c, which is consistent with the value previously reported [26], further confirming that the obtained Sb is present as few-layered Sb nanosheets.

Raman spectroscopy can provide fingerprint spectra of molecular vibrations and rotations. To further characterize the prepared samples, Raman spectra were recorded as shown in Figure 2d. Two characteristic peaks at ~148 cm^−1^ and ~111 cm^−1^ were observed for 2D Sb/TNRAs and attributed to the A_1g_ and E_g_ modes of Sb, respectively [48]. These peaks are blue-shifted compared with bulk antimony, confirming the successful loading of 2D Sb onto TiO_2_ NRAs and the strong interaction between the 2D Sb and TiO_2_ nanorods [26]. The characteristic Raman modes of rutile TiO_2_ are also observed in the synthesized 2D Sb/TNRAs nanocomposites at ~608 cm^−1^, ~445 cm^−1^, and ~240 cm^−1^. The peak at ~240 cm^−1^ can be assigned to the second-order effect (SOE). The Raman results further indicate that 2D Sb/TNRAs heterostructures have been successfully synthesized.

The surface chemical composition of the prepared 2D Sb (4 mg)/TNRAs nanocomposites were analyzed using XPS with different etching depths (Figure 3). Figure 3a shows high-resolution Ti 2p XPS spectra for the nanocomposites unetched and etched 100 and 200 nm. For the 2D Sb (4 mg)/TNRAs nanocomposites etched 100 nm, the XPS spectra show two peaks at 452.8 and 458.5 eV, corresponding to Ti 2p_3/2_ and Ti 2p_1/2_, respectively, which is a typical characteristic of Ti^4+^-O bond in titanium dioxide [55]. When further etching to 200 nm, the peak value of Ti 2p remains almost unchanged. While for the unetched 2D Sb (4 mg)/TNRAs, the binding energy of Ti 2p occurs at 452.6 and 458.3 eV, respectively, which is reduced by about 0.2 eV compared to the pristine TiO_2_ NRAs. The negative shift is due to the electron transfer of 2D Sb to Ti^4+^-O, suggesting the successful loading of 2D Sb onto TiO_2_. The same results were observed from the fine XPS spectra of O 1s with different degrees of etching. As shown in Figure 3b, a negative shift of the binding energies is also observed for the O 1s of unetched 2D Sb (4 mg)/TNRAs compared with those of etched binding energies. Additionally, a new small peak near 528.4 eV was observed for the unetched 2D Sb (4 mg)/TNRAs, which can be assigned to the Sb 3d_5/2_ of metallic Sb (Sb^0^), further indicating the successful loading of Sb nanosheets [26]. According to the XPS results, the thickness of the loaded 2D Sb is about 100–200 nm, which is consistent with the SEM results (Figure 1d).

The UV-Vis absorption spectra of pristine TiO_2_ NRAs and 2D Sb (4 mg)/TiO_2_ NRAs nanocomposites are shown in Figure 4a. The 2D Sb (4 mg)/TiO_2_ NRAs nanocomposites show stronger light absorption than that of pristine TiO_2_ NRAs in the UV light to visible light range. Figure 4b shows the Tauc plots obtained from Figure 4a. The absorption edge of the pristine TiO_2_ NRAs (black line) is at 404 nm, which indicates the energy band gap is 3.06 eV. After loading 2D Sb, the absorption edge of the 2D Sb/TNRAs was shifted to 455 nm, indicating that the combination of 2D Sb with TiO_2_ NRAs could expand the absorption range of the photoelectrode. As can be seen from the Tauc plot, the band gap of the 2D Sb (4 mg)/TNRAs is about 2.72 eV, lower than that of the pristine TiO_2_ NRAs with the band gap reduced by about 0.34 eV. The results demonstrate that the introduction of 2D Sb into TiO_2_ NRAs can broaden the range of the photoresponse, which can result in an enhancement in the utilization of solar light and improve PEC performance. In addition, the enhanced light absorption can be attributed to the existence of few-layered antimonene nanosheets, which are proved to be semiconductors with tunable bandgaps. 

### 3.2. PEC Performance of 2D Sb/TiO_2_ NRAs

The effect of loading 2D Sb onto TNRAs has been investigated on the PEC performances of the 2D Sb/TNRAs nanocomposites through a number of experiments. Figure 5a shows the photocurrent density versus time curves of bare TiO_2_ and 2D Sb/TiO_2_ photoanodes with various 2D Sb loadings. The average photocurrent density of pristine TNRAs, 2D Sb (2 mg)/TNRAs, 2D Sb (4 mg)/TNRAs, and 2D Sb (6 mg)/TNRAs electrodes are found to be ~0.80, ~1.20, ~1.32, and ~0.88 mA/cm^2^, respectively, at 1.23 V vs. RHE. It can be seen that the 2D Sb (4 mg)/TNRAs material exhibits the highest photocurrent density under the same conditions, which is ~1.65 times higher than that of the pristine TiO_2_ NRAs photoanode. When further decreasing or increasing the loaded 2D Sb content, a deterioration in PEC performance occurs for 2D Sb (2 mg)/TNRAs and 2D Sb (6 mg)/TNRAs, respectively, as shown in Figure 5a. For the pure 2D Sb photoanode, the photocurrent density is very weak, only ~2.2 μA cm^−2^, which is consistent with the previous report [42]. As shown in Figure 5b, the photocurrent density increases with increasing applied bias voltage, and the photocurrent density of the 2D Sb (4 mg)/TNRAs is significantly higher than that of the original TNRAs. This enhancement is mainly due to the photogenerated electron-hole pairs becoming separated at the interface of the 2D Sb (4 mg)/TiO_2_ heterojunction and the electron transport along the TiO_2_ nanorods to the FTO substrate so that the charge separation efficiency is promoted. Moreover, the introduction of 2D Sb extends the absorption range to the visible light region. 

To further understand the photo-generated carriers’ transition, open-circuit potential (OCP) measurements were obtained to identify long-lived carriers on a heterogeneous structure generated between the optical collector layer and the buffer layer. Figure 5c shows the OCP curve of the 2D Sb (4 mg)/TNRAs photoanode. Compared to the TNRAs photoanode, the 2D Sb (4 mg)/TNRAs photoanode shows a higher photo-potential value, indicating a higher capability of separating electron-hole pairs with a lower recombination rate. The enhanced photo-potential value is due to the formed heterostructure between the 2D Sb and TNRAs, which facilitates carrier separation [56,57]. 

### 3.3. Possible Mechanism of the Enhanced PEC Performance

To gain a clear idea of the charge-transfer process in the pristine TNRAs and 2D Sb/TNRAs photoelectrodes, an EIS analysis was carried out under simulated solar light illumination. Figure 5d shows the Nyquist plots of the pristine TNRAs, 2D Sb (2 mg)/TNRAs, 2D Sb (4 mg)/TNRAs, and 2D Sb (6 mg)/TNRAs. The smaller semicircle at high frequencies for 2D Sb/TNRAs represents lower charge-transfer resistance, namely, the higher separation efficiency of the photogenerated electron-hole pairs than that of pristine TNRAs. Furthermore, The 2D Sb (4 mg)/TNRAs electrode presents the smallest semicircle at high frequencies among the investigated 2D Sb/TNRAs electrodes with three different 2D Sb loading amounts. Combined with the UV-Vis absorption spectra, it can be concluded that the higher PEC performance of 2D Sb (4 mg)/TNRAs is mainly ascribed to the enhanced light absorption and lowered charge transfer resistance in the interface between the 2D Sb and TiO_2_ NRAs.

According to the results discussed above, a possible mechanism for the charge transfer process in a 2D Sb/TNRAs hybrid system is proposed. As shown in Figure 6, under illumination conditions, when the energy of the incident light is greater than the energy of the semiconductor bandgap, photoelectrons and holes are generated in the conduction band (CB) and the valence band (VB) of the TNRAs and 2D Sb, respectively. Compared with TiO_2,_ the CB of 2D Sb is more negative, so the photo-generated electrons in 2D Sb can be easily transferred to the TiO_2_ nanorods. The ordered 1D structure of the TiO_2_ NRAs provides a conduction path for electrons and rapidly transfers electrons to the FTO along the vertically oriented TiO_2_ nanorods, which finally reach the counter electrode for hydrogen generation. At the same time, these holes left in the VB of TiO_2_ are transported to the VB of 2D Sb and oxidized water to produce O_2_. The migration of electrons and holes in opposite directions increases the efficiency of the charge separation and effectively improves the PEC performance.

Figure 7 compares this work with related work reported in recent years to improve the PEC performance by constructing heterojunctions [58,59,60,61,62,63]. Comparing the photocurrent density at 1.23 V vs. RHE, the 2D Sb/TNRAs photoanode reported in this work shows a comparatively high photocurrent density, indicating that the synthesized 2D Sb/TNRAs composites show certain potential in photoelectrocatalytic applications such as for water splitting.

## 4. Conclusions

In summary, we designed heterostructured 2D Sb/TiO_2_ nanorod arrays via a facile drop-coating method from liquid-exfoliated 2D Sb suspensions and investigated their PEC performances for water splitting. As a photoanode for PEC water splitting, the optimal 2D Sb/TNRAs photoanode showed a significantly enhanced photocurrent density (1.32 mA/cm^2^, AM 1.5 G), which is 1.65 times higher than that of pristine TiO_2_ nanorod arrays. We conducted an in-depth study on the enhancement of PEC performance and proposed a possible mechanism. The troublesome charge-transfer resistance of bare TiO_2_, which blocks the PEC activity, was significantly reduced after the introduction of 2D Sb onto TiO_2_ NRAs. The 2D Sb/TNRAs photoanode shows higher electron-hole separation efficiency and lower interfacial impedance. Moreover, it can be observed from UV-Vis absorption spectra that the 2D Sb/TNRAs heterostructure has an enhanced absorption intensity throughout the ultraviolet to visible light region. Therefore, the construction of 2D Sb/TNRAs heterojunctions plays an important role in inhibiting the photogenerated electron-hole pair recombination and increasing the charge transfer rate.

## Figures and Tables

**Figure 1 nanomaterials-13-01293-f001:**
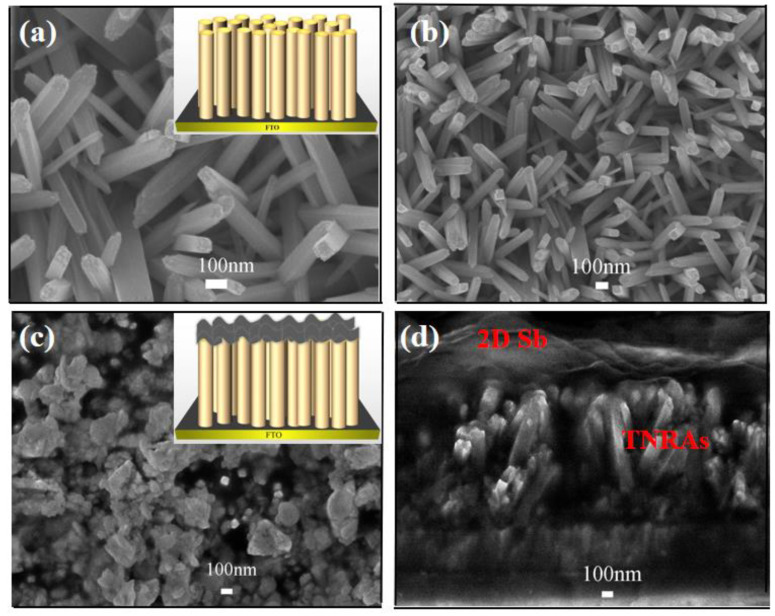
(**a**,**b**) Top view SEM of TiO_2_ NRAs. (**c**,**d**) SEM images of 2D Sb/TiO_2_ NRAs: (**c**) top view, (**d**) cross-sectional view.

**Figure 2 nanomaterials-13-01293-f002:**
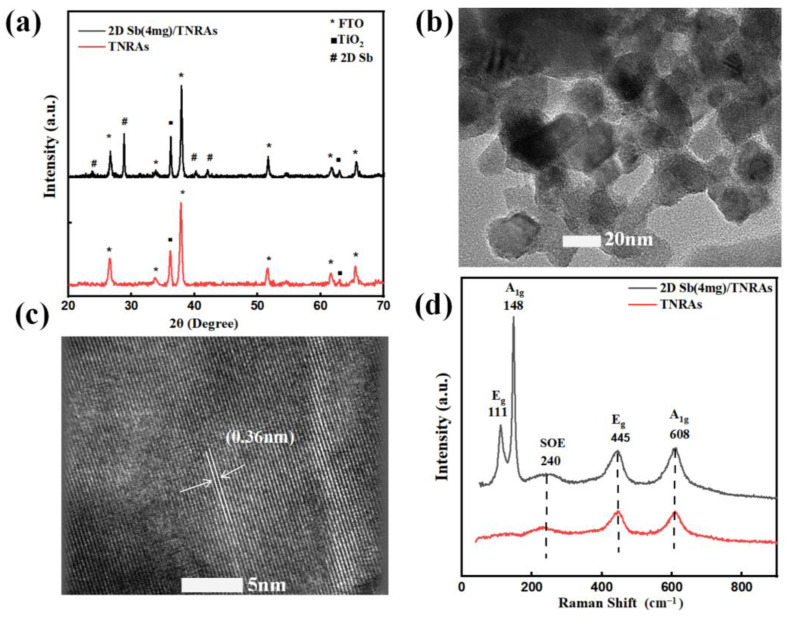
(**a**) XRD patterns and (**d**) Raman spectra of 2D Sb/TiO_2_ NRAs. TNRAs: TiO_2_ nanorod arrays. (**b**) TEM and (**c**) HRTEM images of obtained 2D Sb.

**Figure 3 nanomaterials-13-01293-f003:**
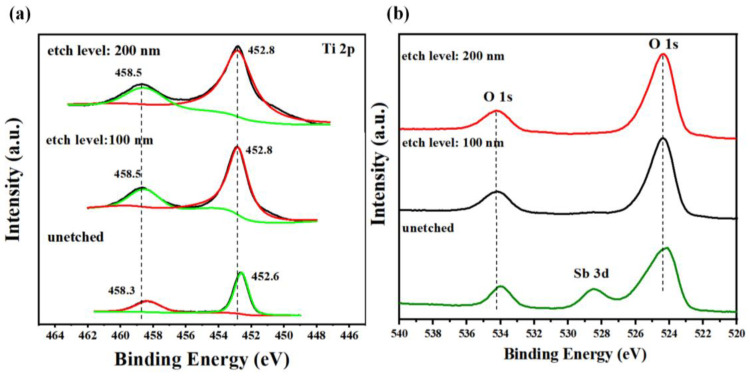
Fine XPS spectra of 2D Sb (4 mg)/TiO_2_ NRAs with different etching depths. (**a**) Ti 2p. (**b**) Sb 3d and O 1s. TNRAs: TiO_2_ nanorod arrays.

**Figure 4 nanomaterials-13-01293-f004:**
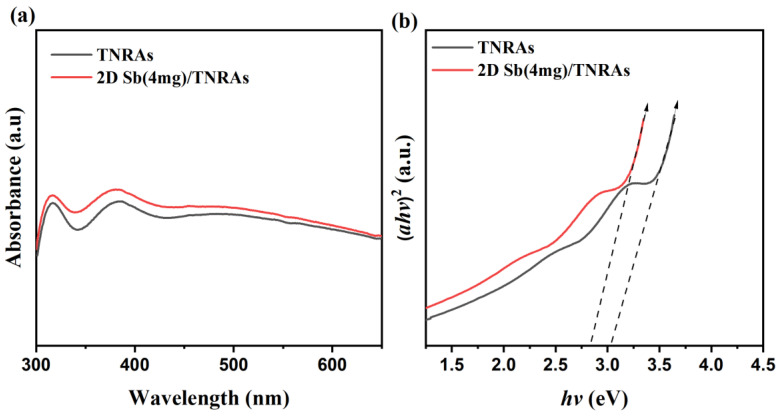
(**a**) UV-Vis absorption spectra of TiO_2_ NRAs and 2D Sb (4 mg)/TiO_2_ NRAs. (**b**) Tauc plots from the UV–Vis spectra of TiO_2_ NRAs and 2D Sb (4 mg)/TiO_2_ NRAs. TNRAs: TiO_2_ NRAs.

**Figure 5 nanomaterials-13-01293-f005:**
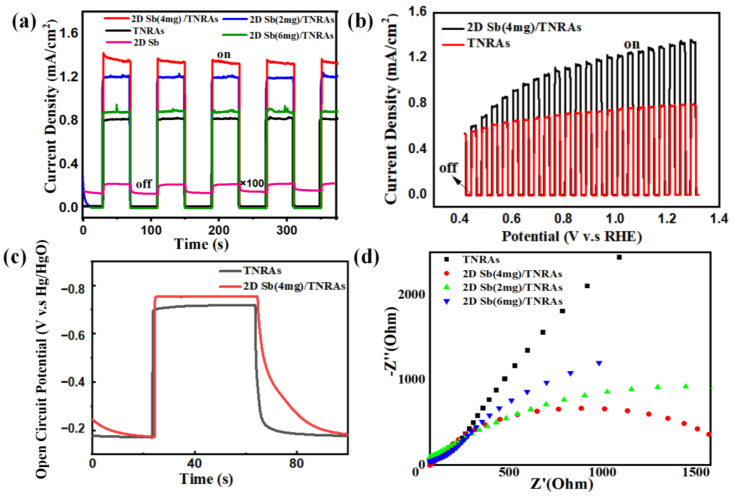
The PEC performance of the synthesized 2D Sb/TNRAs photoanodes with different 2D Sb loading measured in 1 M KOH under AM 1.5 G irradiation. (**a**) Transient photocurrent responses collected at 1.23 V vs. RHE. (The photocurrent density of 2D Sb is amplified 100 times for comparison). (**b**) LSV curves. (**c**) Open-circuit potential curves. (**d**) EIS spectra. TNRAs: TiO_2_ NRAs; 2D Sb (4 mg)/TNRAs: 2D Sb (4 mg)/TiO_2_ nanorod arrays.

**Figure 6 nanomaterials-13-01293-f006:**
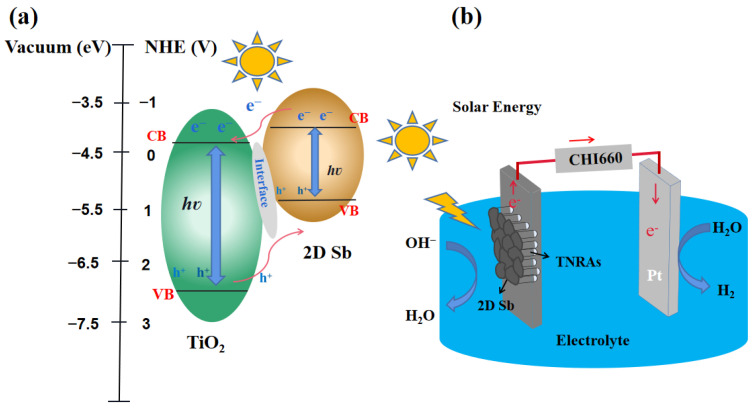
(**a**) Schematic diagram for the separation of photogenerated electron-hole pairs and the charge transfer process in a 2D Sb/TNRAs heterojunction system. (**b**) Schematic illustration of photoelectrochemical water-splitting process over the 2D Sb/TNRAs system. Conduction band (CB) and valence band (VB) edges for TiO_2_ and 2D Sb were taken from the literature [24,26].

**Figure 7 nanomaterials-13-01293-f007:**
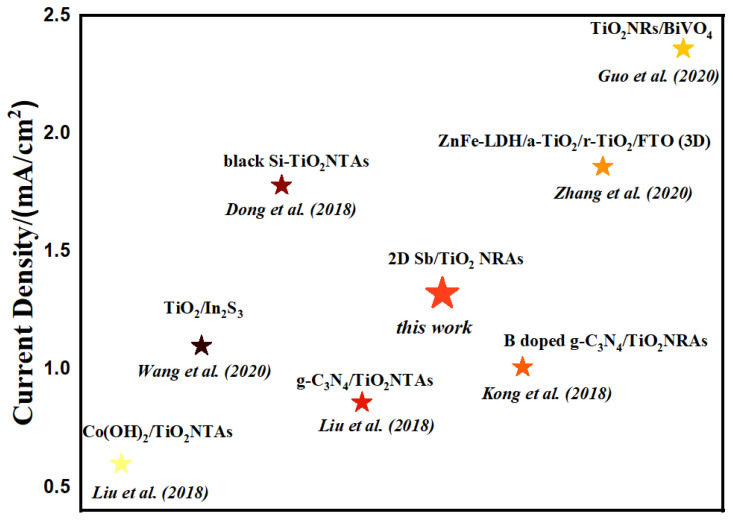
Comparison chart demonstrating the photocurrent density of previous works and this work. Each photocurrent density was obtained at 1.23 V vs. RHE [23,58,59,60,61,62,63].

## Data Availability

The data is available on reasonable request from the corresponding author.

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
