# Peer review of "Two-Dimensional Sb Modified TiO2 Nanorod Arrays as Photoanodes for Efficient Solar Water Splitting"

_nanomaterials, 2023, doi:10.3390/nano13071293_

Round 1

Reviewer 1 Report

The present manuscript described the photocatalytic watersplitting efficiency of TiO2-Sb.

This manuscript lacks sufficient novelty, since a numerous literature available for the  TiO2 based  photocatalytic watersplitting applications.

The only novelty stands on 2D Sb based composite.

This work needs more recent literature update on TiO2-Sb based materials. 

The proposed reaction mechanism is not acceptable at this stage. Since SB being the metal, it doesn't have bandbag energy. But in the proposed reaction mechanism, it was reported the band gap energy, discussing the  photo sensitive effect. Which is controversy. Please justify this mechanism in a more concise way. 

The citation numbering needs to fit the journals guidelines..

The grammatical errors needs to be corrected throughout the manuscript.

Author Response

Thank you for your letter and for the reviewer 1’s comments concerning our manuscript entitled “Two-dimensional Sb modified TiO2 nanorod arrays as photoanodes for efficient solar water splitting” (nanomaterials-2266392). Those comments are all valuable and very helpful for revising and improving our paper, as well as the important guiding significance to our researches. We have studied comments carefully and have made correction which we hope meet with approval. Revised portion are marked in highlight yellow in the paper. The detailed reply is in the attached file. Please check it. Thank you.

Reviewer 2 Report

The authors' report on the Two-dimensional Sb modified TiO2 nanorod arrays as photoanodes for efficient solar water splitting. The manuscript was poorly written. I recommend that the authors address the following queries before acceptance:

1.      Why did the authors select a particular amount (100 μL) of 2D Sb suspension? Authors must study the variation of concentration of Sb solution to optimize the PEC performance and provide a detailed explanation in the manuscript.

2.      Authors should provide the TEM images of pristine 2D Sb nanosheets to confirm the morphology of the synthesized material.

3.      Authors must check Figure 2a. Why is there a representation of SnO2? This needs to be addressed and corrected.

4.      Why are there no XPS details for Sb? Authors should provide XPS data to confirm the presence and oxidation state of Sb in the synthesized material.

5.      Why did the authors select drop casting over other methods like spin coating for making Sb modified TiO2 nanorod arrays? This needs to be explained in the manuscript.

6.      Authors shown Figure 3b as O 1s XPS but written in text as Sb XPS analysis. This needs to be corrected.

7.      Additionally, authors must cite literature related to metal oxide-related PEC articles in the introduction appropriately. Chemosphere, 305 (2022) 135461. Materials Letters, 303 (2022) 131812. Chemosphere, 286 (2022) 131577. Materials Letters, 274 (2020) 128005. Journal of Electroanalytical Chemistry, 900 (2021) 115699.

Author Response

Thank you for your letter and for the reviewer 2’s comments concerning our manuscript entitled “Two-dimensional Sb modified TiO2 nanorod arrays as photoanodes for efficient solar water splitting” (nanomaterials-2266392). Those comments are all valuable and very helpful for revising and improving our paper, as well as the important guiding significance to our researches. We have studied comments carefully and have made correction which we hope meet with approval. Revised portion are marked in highlight yellow in the paper. The detailed reply is in the attached file. Please check it. Thank you.

Reviewer 3 Report

This manuscript presents a high scientific level study of heterostructures constituted of TiO2 films coated with 2D antimonene films. The characterization section including XRD, Raman and SEM analyses is convincing. The SEM images are convincing and clearly suggest the formation of heterostructures. The resulting properties are correctly described and discussed.

Author Response

Thank you for your letter and for the reviewer 3’s comments concerning our manuscript entitled “Two-dimensional Sb modified TiO2 nanorod arrays as photoanodes for efficient solar water splitting” (nanomaterials-2266392). Those comments are all valuable and very helpful for revising and improving our paper, as well as the important guiding significance to our researches. We have studied comments carefully and have made correction which we hope meet with approval. Revised portion are marked in highlight yellow in the paper. The detailed reply is in the attached file. Please check it. Thank you.

Reviewer 4 Report

The article is interesting and well presented. However, I would like to suggest the following to the authors:

- to comply with the guide for editing the paper according to the journal's instructions, format, references, etc.

- to reorganise the Result and discussion part, also specifying the types of characterisations carried out;

- to revise the English language.

Author Response

Thank you for your letter and for the reviewer 4’s comments concerning our manuscript entitled “Two-dimensional Sb modified TiO2 nanorod arrays as photoanodes for efficient solar water splitting” (nanomaterials-2266392). Those comments are all valuable and very helpful for revising and improving our paper, as well as the important guiding significance to our researches. We have studied comments carefully and have made correction which we hope meet with approval. Revised portion are marked in highlight yellow in the paper. The detail reply is in the attached file. Please check it. Thank you.

Round 2

Reviewer 1 Report

The revised manuscript is now acceptable for publication in the journal.

Reviewer 2 Report

The authors have done the revision satisfactorily and answered all the questions raised.

The current version of the manuscript can be accepted.